# Patient-Derived Bladder Cancer Organoid Models in Tumor Biology and Drug Testing: A Systematic Review

**DOI:** 10.3390/cancers14092062

**Published:** 2022-04-20

**Authors:** Benjamin Medle, Gottfrid Sjödahl, Pontus Eriksson, Fredrik Liedberg, Mattias Höglund, Carina Bernardo

**Affiliations:** 1Division of Oncology, Department of Clinical Sciences Lund, Lund University, Medicon Village, Scheelevägen 2, 223 81 Lund, Sweden; b.medle@gmail.com (B.M.); pontus.eriksson@med.lu.se (P.E.); mattias.hoglund@med.lu.se (M.H.); 2Division of Clinical and Experimental Urothelial Carcinoma Research, Department of Translational Medicine, Lund University, Malmö and Department of Urology, Skåne University Hospital, Jan Waldenströms Gata 5, 205 02 Malmö, Sweden; gottfrid.sjodahl@med.lu.se (G.S.); fredrik.liedberg@med.lu.se (F.L.)

**Keywords:** bladder cancer, organoids, spheroids, precision medicine, 3D tumor models, drug response

## Abstract

**Simple Summary:**

Primary culture of cancer cells from patient tumors in a physiologically relevant system can provide information about tumor biology, disentangle the role of different cell types within the tumors, and give information about drug sensitivity for the development of cancer-targeted therapies and precision medicine. This requires the use of well-characterized and easily expandable tumor models. This review focuses on 3D models developed from primary human tissue including normal urothelium or bladder cancer samples, the characteristics of the models, and to what extent the organoids represent the diversity observed among human tumors.

**Abstract:**

Bladder cancer is a common and highly heterogeneous malignancy with a relatively poor outcome. Patient-derived tumor organoid cultures have emerged as a preclinical model with improved biomimicity. However, the impact of the different methods being used in the composition and dynamics of the models remains unknown. This study aims to systematically review the literature regarding patient-derived organoid models for normal and cancer tissue of the bladder, and their current and potential future applications for tumor biology studies and drug testing. A PRISMA-compliant systematic review of the PubMED, Embase, Web of Sciences, and Scopus databases was performed. The results were analyzed based on the methodologies, comparison with primary tumors, functional analysis, and chemotherapy and immunotherapy testing. The literature search identified 536 articles, 24 of which met the inclusion criteria. Bladder cancer organoid models have been increasingly used for tumor biology studies and drug screening. Despite the heterogeneity between methods, organoids and primary tissues showed high genetic and phenotypic concordance. Organoid sensitivity to chemotherapy matched the response in patient-derived xenograft (PDX) models and predicted response based on clinical and mutation data. Advances in bioengineering technology, such as microfluidic devices, bioprinters, and imaging, are likely to further standardize and expand the use of organoids.

## 1. Introduction

Bladder cancer is a common and highly heterogeneous malignancy that manifests in two major patterns: As non-muscle invasive tumors (NMIBC), which make up around 75% of the new cases and generally have a better prognosis but frequent relapses; or as muscle invasive tumors (MIBC) with high risk of regional and distant metastasis and poor prognosis [1]. Patients with NMIBC can be treated with tumor resection (transurethral resection of the bladder (TURB)) and intravesical therapy, but up to 30% of the patients will progress to MIBC, requiring surgery and/or systemic treatment. At this stage, despite treatment, five-year overall survival is only 50% [2]. Regardless of the advancing pace of molecular characterization and therapeutic targeting of many cancer types, bladder cancer is lagging behind, with few improvements to clinical management and disease outcome over the past decades; this is mainly related to disease heterogeneity and the absence of targeted therapies. The development of models that faithfully recapitulate the biology and complexity of these tumors in an in vitro system, amenable to manipulation, has also been a major challenge.

Three-dimensional (3D) organoids have become a powerful tool to study the molecular and cellular basis of epithelial differentiation. Organoids have been defined in different ways in various subfields. In cancer research, organoids have been defined as structures containing several cell types that develop from cells capable of self-renewal and self-organization through cell sorting and lineage commitment similar to the process vivo [3]. Furthermore, organoids can refer to clonal derivatives of primary epithelial stem cells grown without mesenchyme or to epithelial-mesenchymal co-cultures [4]. These models preserve the 3D organization of the tissues, retain architecture as well as cell–cell and cell–matrix interactions, features mainly lost in 2D cultures. Organoids have proven to be a powerful tool to study tissue morphogenesis, developmental biology, cancer heterogeneity, and drug screening, and for establishing a solid basis for regenerative medicine and gene therapy [5].

The development of multicellular tumor models was introduced in the early 1970s by radiobiologists using cancer cell lines to form spheroids [6]. Over the years, other types of 3D models have been reported. Although 3D morphology is a shared property, the nomenclature varies a lot and leads to some confusion. From a methodological perspective, spherical cancer models can be classified into four groups [7]; two of them are derived from single cell suspensions and two derived from tumor tissue: (1) multicellular tumor spheroids, formed in nonadherent conditions from single-cell suspension; (2) tumorspheres developed from proliferation of cancer stem cells (CSC) in serum-free medium supplemented with growth factors; (3) cancer tissue originated spheroids (CTOS), formed after partial dissociation of the tumor tissue; (4) organotypic multicellular organoids, generated after cutting the tumor tissue into small pieces. More recently, the term assembloid, which was originally used to describe neural 3D structures formed from the fusion and functional integration of multiple cell types [8], has been used to describe tumor organoids assembled with stromal cells. Each of these main groups are characterized by differences in sample preparation, cell density, medium composition and handling. Over time they have been described in the literature using this terminology as well as additional alternative terminologies. For simplicity, we will refer to organoids as an overarching term for all these models or use the terminology in the original publication.

One of the early key events in the development of the tumor organoid field as we know it today occurred more than thirty years ago when Mina Bissel and her research group developed cultures in laminin-rich gels and elucidated the effect of extracellular matrix on breast cancer gene expression [9,10]. More recently, Clevers’ research group has established intestinal organoids with crypt–villous structures [11]. The method was then expanded to other organs and different tumor types, including liver, prostate, lung, and pancreas [12,13,14]. In cancer research, this approach has been used to identify and culture cancer stem cells as well as for functional assays and drug testing. Tumor organoids have been established from regular cancer cell lines, human tumor surgical specimens, and patient-derived xenograft models (PDX). These organoids can be generated from single cells, cell clusters, or tumor fragments.

Bladder cancer organoids have been recently described by several groups [15,16,17,18,19] as a tool for studying molecular tumor characteristics and cell dynamics. In the early years of bladder cancer organoids, many of the studies used established cancer cell lines as a source for the organoids, as reviewed by Vasyutin et al. [20]. The number of publications, size of the cohorts, and depth of the molecular analysis has increased during recent years. Still, there is a lack of a comprehensive evaluation of the different methods being used and how they impact the composition and dynamics of the organoid models. Additionally, different approaches to sample processing and culture methodology might select for different cell populations or generate models representative of different tumor subtypes. Considering the potential use of organoids for functional assays and as a platform for drug screening to inform therapeutic decisions, a systematic and critical review of the existing literature is lacking, not only to understand the available evidence but also to improve future studies and reproducibility.

This review analysis focuses on 3D models developed from primary human tissue including normal urothelium or bladder cancer samples, the characteristics of the models, and to what extent the bladder cancer organoids represent human tumor diversity observed in the clinical setting. Thus, the aim of this systematic review is to provide a detailed description of the procedures currently used to establish and characterize tissue-derived human bladder cancer organoids and their application as preclinical models.

## 2. Materials and Methods

The preregistered protocol for this systematic review is available at Open Science Framework (https://osf.io/, protocol ID reqmf, registered on 14 February 2022).

### 2.1. Search Strategy and Eligibility Criteria

Two authors (BM and CB) searched the electronic databases of PubMed, Embase, Web of Science, and Scopus to identify studies using tissue-derived human bladder cancer organoids published between January 2000 and November 2021. Search queries were developed through an iterative process for each database to identify reports of 3D cultures derived from normal or bladder cancer human tissue. The full search strategy is described in Appendix A. Included studies were limited to original publications in English. References of included studies and relevant reviews were searched for additional studies.

### 2.2. Study Selection

Two reviewers independently performed the abstract screening, full text assessment, and data extraction. Discrepancies were addressed by discussion and eventual consultation with a third reviewer. The inclusion criteria consisted of use of primary human tissue including normal urothelium or bladder cancer samples as source material to establish 3D cultures; both clinical samples and patient-derived xenografts (PDX) were accepted. Studies using established cell lines or tissue from non-human origin were excluded. When multiple reports were derived from the same cohort of samples, the most complete report was considered. Reporting of essential information, such as the primary samples used and methods for sample processing and culture, was criteria for inclusion. Only original articles were included.

### 2.3. Data Extraction and Analysis

Data were extracted by two reviewers independently from the full-text publications using a predefined data collection form. Data extracted included publication information, sample characteristics and processing, culture conditions, experimental conditions, and analysis. The results were analyzed based on the methodologies, comparison with primary tumors, functional analysis, and chemotherapy and immunotherapy testing. In cases where the information provided for a specific parameter was not clear or enough to analyze the data, the report was not included in the analysis.

## 3. Results

The literature search yielded 534 articles, with 2 additional reports found after screening of reference lists. After removal of duplicates, 245 records were screened, narrowing down to 31 which underwent full review, and ending with 24 included in the final analysis (Figure 1). One study applied machine learning to previously published data [21] and is considered as the same cohort of samples as the original/source study [15]. Appendix A includes detailed information for each study.

### 3.1. Characteristics of Included Studies

Among the studies included, 3 used normal urothelium as starting material and 23 used human tumor samples, of which 13 specimens were directly from TURB or cystectomy, 4 from PDX models, 6 from both patient samples or PDX models, and 1 from urine sediments (Table 1). The selected studies originated from the USA (*n* = 10), eastern Asia (*n* = 9), and Europe (*n* = 5). The main purposes of the studies included characterization of the organoids, drug testing, and investigation of bladder cancer stem cells and tumor heterogeneity. Three papers reported data obtained from the same cohort of spheroids established from an expanding cohort of tumor samples. There was not enough information to distinguish overlapping from newly established models, and thus these papers were treated as one cohort [22,23,24]. The number of organoid models described in each study varied from 2 to 128, with a total of 344 individual models identified. The organoids were obtained with diverse methods, exposed to different growth conditions, treatments, co-cultures, and genetic manipulations. Most studies reported the establishment of organoids from urothelial carcinoma and both non-muscle invasive (NMIBC) and muscle-invasive bladder cancers (MIBC), with low-grade and high-grade samples. In addition to pure urothelial carcinoma (UC), squamous cell carcinoma, UC with divergent differentiation, and UC with concomitant carcinoma in situ (CIS) were also represented in the organoids originating from patient samples [15,17]. Additionally, one study reported establishment of organoids from two neuroendocrine bladder tumors [25]. Information regarding primary tumor histology was missing in only three studies [26,27,28]. However, in the remaining reports, the proportion of successful organoids from each group of samples was not always disclosed.

### 3.2. Quality of Design and Reporting

Three studies were excluded during the selection process due to lack of information regarding methods used [42,43,44]. Most papers included the basic information about sample selection, sample processing, and culture conditions. Information about patient treatment status and tumor stage and grade for the samples was often missing and thus not analyzed in this review. Some reports provided scant information about organoid model use and characterization, referring to them only as a complementary system in their study [26]. These reports were included in the analysis of the methodology, but not further analyzed in terms of applications and results.

### 3.3. Methodology for Tumor Dissociation and Organoid Culture

Figure 2 summarizes the different methods used for sample processing and culture. Two main approaches were used in the establishment of bladder cancer organoids: small cell clusters or single cells were grown either in floating aggregation culture or embedded in laminin-rich extracellular matrix (ECM) gels. The most common culture media consist of DMEM/F-12 medium or Advanced DMEM/F-12 supplemented with growth factors and differentiation and cell death inhibitors (Table 2). RPMI and hepatocyte medium supplemented with fetal bovine serum (FBS) or different combinations of growth factors were also successfully used. These are media formulations optimized for the growth and expansion of human stem cells or other mammalian cells in serum- and feeder-free conditions.

There is a great variety in the extent and duration of primary tissue dissociation before culture (Table 2). The methods used in bladder cancer reflect the different methodologies developed over time for 3D systems [7,45]. An early study [29] used a method similar to the explant model as described by Bjerkvig et al., which consisted of simply cutting the cancer tissues into 0.3–0.5-mm pieces and culturing it in agar-coated tissue flasks with media supplemented with FBS and an excess of non-essential amino acids [46]. More recently, a similar approach was used to establish air–liquid interface organoids from a large cohort of tumor samples, including at least one from bladder cancer [27]. In this report, the samples were finely minced, resuspended in Collagen I and layered on top of pre-solidified collagen gel. The Transwell was then moved to a cell culture dish containing stem cell medium.

In a second approach, bladder cancer spheroids are established after partial dissociation of the tumor tissue [19,22,23,24,26,30,32,40], maintaining cell–cell contact of cancer cells as described by Kondo et al. for colorectal cancer spheroids [47]. The cultures derived from this method are often termed cancer-tissue originated spheroids (CTOS). The enzymes and duration of the dissociation vary between studies (25–120 min). The tissue fragments retained on the strainers are used for organoid culture, while the flow-through and the cells in the supernatant fraction are discarded. The cell clusters are then cultured with stem cell medium (without FBS and supplemented with growth factors) in suspension and, in some cases, transferred to a matrix such as Cellmatrix, collagen, Matrigel, or BME after 24 h (Table 2). This approach has also been coupled with culture in microchambers to facilitate gas and nutrient exchange as well as diffusion of drugs [32].

A third method consists of partial or full dissociation of the tumors, culturing single cells and smaller cell clusters which passed through the strainers [15,16,17,25,31,37,38,39,41] or only single cells [28]. After dissociation, the cells are cultured with or without matrix in stem cell medium or FBS-containing medium. In the anchorage-independent approach, the free-floating tumorspheres form in low-adherence conditions that promote cell–cell adhesion and formation of aggregates. When embedded in a cellular matrix, right after dissociation, single cells can either still come together to form multiclonal structures or form organoids from individual clones. The most common approach is to use 50% of cellular matrix or higher for embedding, but a concentration as low as 5% or even 2% Matrigel has also been used [28,39].

#### Co-Culture and Bioprinting

Within the timeframe of this review, the first organoid models of bladder cancer consisting of epithelial and stromal cells were established by plating mechanically dissociated tumor tissues in a type I collagen matrix [27]. Subsequently, Kim et al. [37] developed a complex system to assemble three-layered normal bladder organoids, assembloids, by adding stromal cells such as normal fibroblasts, endothelial cells, and human induced pluripotent stem cell (hiPSC)-derived smooth muscle cells to urothelial cells in a bioreactor [37]. This method was also used for tumor organoids, where cancer-associated fibroblasts (CAFs) and endothelial cells of human origin (HULECs) were added to bladder cancer cells. The process to produce these assembloids was later automated using high-throughput 3D bioprinting.

Another alternative approach to model the immune response in vitro was described by Yu et al., using a co-culture of tumor organoids with T cells genetically engineered to express chimeric antigen receptors (CAR) targeting a tumor-associated antigen [41].

Bioprinting was also used to quantify intratumoral heterogeneity from tumor organoids [28]. After the initial establishment of organoids from dissociated tumor samples using the hanging drop method in Matrigel and stem cell media, the organoids were dissociated into single cells, supplemented with 2% Matrigel, and inkjet printed on ultralow-adhesion 384-well plates. Individual cells grown into organoids were further expanded for analysis. This study shows that tumor cells remain viable after printing, tumor organoids can be generated from single cells, and that organoids derived from individual tumor samples can present heterogeneity in terms of growth rate, gene expression, and drug sensitivity [28].

### 3.4. Efficiency of Tissue-Derived Organoid Production

Reported success rates in the establishment of organoids ranged between 33–100% (Table 1). The criteria to define a successful establishment varied between studies, mainly in terms of number of organoids (from 3 to 72) obtained and suitability for drug testing [22,23,29,36], survival of the organoids for more than one week [31], or even successful serial passage at least six times [15,33]. In the latter case, the authors noted that some organoid lines fail to propagate after the first three to five passages, where the success rate drops from 100% to 70% for long-term cultures.

Organoids derived from normal urothelium were obtained in close to 50% of the 39 samples [17] and in 6 out of 6 samples in an earlier report [30]. For tumor samples, reported success was found higher among NMI tumors (ranging from 50–90.7%) when compared to MI samples (20–68%) [23,36]. However, another study analyzing almost one hundred samples reported growth in 60–70% of the cases, without major differences between tumor stage [17]. Of note, organoids established from PDX models had a reported success rate of 100% both for neuroendocrine bladder cancer (NEBC, two samples) [25] and UC (6/6) [16], while in the latter study they had no success with direct patient samples.

### 3.5. Applications

#### 3.5.1. Comparison with Primary Tumors

In comparative analysis, the organoids faithfully recapitulated parental tumor morphological and genetic features [15,16,17,35,37,41]. Morphologic appearance was described as round or spheroid, solid or lumen-containing, and with smooth or irregular borders [17]. Below we summarize the main findings regarding cellular composition, tumor cell phenotypes, and genetic profiles.

Organoids established after mechanical cutting of the primary tissue without dissociation preserved integrated stroma expressing vimentin and SMA [27,29]. The presence of these cells was not observed in methods using enzymatic dissociation, regardless of the extent of dissociation. Organoids derived from both clinical samples and PDX model were negative for stroma markers, including CD45 and CD31, as shown by immunohistochemistry and transcriptomic analysis [16,22].

The organoids were composed of cells expressing epithelial markers such as EpCAM and E-cadherin [19,22] and could show distinct tumor cell phenotypes. The expression of cell adhesion molecules and integrins was noted both in organoids established from cell clusters and from single cells. A few studies investigated organoid tumor-cell phenotype [15,16,17,35,37,41] and reported high concordance between organoids and primary tumors. Additionally, organoids showed characteristics of basal-like or luminal-like phenotype and consisted of cells with different levels of differentiation. In the large cohort of organoid lines established by Mullenders et al. [17] from different historical tumor types, some organoids were found to express both KRT5 and KRT20 in different cell layers or have a more dominant cell type, while TP63 and CD44 were ubiquitously expressed in both normal and tumor-derived organoids [17]. In some cases, organoid lines obtained from the same patient could have different profiles in terms of differentiation. Different organoid lines with clear luminal or basal-like subtype were confirmed by RT-qPCR. However, whether the subtypes are maintained over time in these cultures was not investigated.

Phenotype stability during culture was investigated in the study by Lee et al., where 22 organoid lines were established from non-invasive and invasive tumors [15]. Immunostaining with a panel of basal and luminal antibodies showed that some (36%) of the organoid lines had strong phenotypic stability, whereas a second group (64%) had changes suggestive of a transition to a basal phenotype when in organoid culture. These changes showed no correlation with pathology, mutation profile, modifications in the variant allele fractions, or drug response. Instead, they often reverted in xenografts, suggestive of cellular plasticity [15]. In a later report, Cai et al. compared an early with a later passage of PDX tumors from one therapy-resistant MIBC and respective organoid lines [16]. Both late-passage PDX and organoid systems had downregulation of genes involved in cell differentiation, tissue development, and cell death. The expression profiles and molecular classification identified the primary tumor, respective PDX tumors, and organoids as basal/squamous subtype (Ba/Sq). However, late passage organoids were assigned to the neuroendocrine-like (NE-like) subtype, reflecting a decrease in the expression of basal keratins and luminal markers as shown in supplementary Figure 2 of their paper. Changes in molecular phenotype were also investigated in the context of co-culture developed by Kim et al. by combining epithelial cells, fibroblasts, and endothelial cells into assembloids [37]. In this study, monocultures of tumor organoids established from MIBC showed that basal organoids maintain their phenotype, but luminal tumor organoids gained basal phenotype over time. This luminal-to-basal shift was prevented in co-culture with CAFs. This platform allowed identification of the FOXA1–BMP–hedgehog axis as key players in molecular switching [37].

In the case of less common histological types such as the NEBC histology, the identity of the organoids was also preserved. In the two cases described by Hofner et al., NEBC samples from PDX models were dissociated and cultured in serum-free medium in ultra-low-adhesion flasks [25]. PDX tumors and spheroids both expressed CD56 and synaptophysin in concordance with the primary tumor. The cultured cells were positive for CD47, CD24, CD147, and MET; negative for CD44, CD87, CD133, and CD26; and retained tumorigenic potential in vivo. Further analysis highlighted the role of hepatocyte growth factor/MET for NEBC growth in vitro and as a potential therapeutic target.

Somatic mutation profiles and chromosomal aberrations were highly conserved between the tissue–organoid pairs and were retained after consecutive passages [15,16,41]. In Lee et al., mutation profiles of organoid lines and respective parental tumors showed high concordance, with more than 80% in 11 organoid lines and only 4 organoid lines with less than 60% [15]. The mutation profiles of the organoid lines presented common genetic alterations in epigenetic regulators frequently found in bladder cancer. Additionally, mutations in genes such as *FGFR3*, *STAG2*, *ERBB2*, *EGFR*, *TP53*, and *RB1* were also observed. Deep sequencing and comparison of mutations in the primary tumor with early and late organoid passages showed that the genotype was largely retained. However, clonal evolution during serial passage and interconversion between organoids and xenografts was also noted [15]. Furthermore, the mutation status of *TP53* and *FGFR3* was correlated with response to MDM2 inhibition and growth factor independency [17].

Overall, the existing data suggest that tumor cell morphology and genetic features are preserved in organoid culture. The gene expression profile of the organoids also resembles the parental tumors in terms of molecular subtype and cellular composition. However, in some instances the gene expression profile changed in culture for some of the organoids, particularly those derived from tumors with luminal phenotype, such changes were reversible. A summary of the molecular analyses used to characterize bladder cancer organoids and main applications of the models is presented in Figure 3.

#### 3.5.2. Functional Studies

In addition to profiling and comparison with the primary tumors, some studies used organoids to investigate specific pathways. As expected, the response to growth factors varied across different models and could be evaluated by supplementing the growth media, knockout and overexpression systems, treatment with small molecules, as well as by co-culture. For successful culture of organoids derived from normal bladder, both FGF7 and FGF10 were found sufficient to allow growth in a media containing noggin, R-spondin, and EGF [17]. In cancer organoid cultures, different responses to FGF withdrawal and nutlin treatment (inhibitor of P53 and MDM2 interaction) were observed according to the genetic background of the tumor cells [17]. HER3 activation was shown to increase proliferation in some models, an effect that could be blocked by PI3K and mTOR inhibitors [22]. Similarly, Yoshida et al. showed that Wnt/β-catenin activation increased proliferation and viability of cancer cells in organoid culture [35]. Changes associated with epithelial-to-mesenchymal transition (EMT) and cell attachment were investigated by Yoshida et al., who reported dynamic changes in ΔNp63α in tumor cells during attachment to the matrix and cadherin switching [24]. The expression of stemness-related markers ALDH1A1 and SOX2 was investigated by Namekawa et al. in two patient-derived organoids established from high-grade stage T1 bladder tumors [38]. ALDH activity modulated proliferation and viability of the organoids, and RAR-mediated transcription of downstream targets, including TUBB3, was revealed as a possible mechanism.

Finally, signaling factors from the stroma, particularly those involved in hedgehog signaling, were shown to increase proliferation of both epithelial and stromal cells when in co-culture [37].

#### 3.5.3. Cancer Stem Cell (CSC) Isolation and Characterization

Successful propagation of organoids over long periods of time requires the capture and expansion of tumor cells with CSC abilities. Several studies showed that it is possible to grow both normal and cancer-derived organoids over extended periods of time [15,17,33]. In addition, three studies used organoids to directly isolate and characterize CSC as well as the conditions for its propagation.

Fierabracci et al. established bladder spheroids from partially dissociated surgical specimens of normal bladder cultured in stem cell medium suspension [30]. Established spheroids were dissociated and subcultured every 7–10 days for two months, after which they were analyzed by FACS. CD34 was found expressed in 1–2.5% of the cells of all six spheroid lines. Some lines also contained cells expressing CD117 or uroplakin II, desmin, and α-SMA.

Additionally, in Bentivegna et al., the authors compared two medium formulations to grow bladder cancer stem-like cell populations from clinical samples subjected to mechanical and enzymatic dissociation [31]. They observed sphere formation in both media but limited proliferation and expansion in the “stem cell media”. The cells in the spheres showed less cytogenetic complexity and chromatin instability than the primary tumors, suggesting a selection during culture. Both spheres and adherent cultures derived from the spheres contained a heterogeneous population of OCT3/4+, CD133+, and nestin+ progenitor cells and a smaller portion of keratin-expressing cells. The cells showed morphological changes suggestive of differentiation when cultured with serum-supplemented media. The tumorigenicity of the isolated CSCs remained to be confirmed.

In a different approach, Ooki et al. started by sorting the cells and comparing the cancer stem cell traits of high-CD24 and low-CD24 tumor cells isolated from PDX models [34]. High-CD24 expressing cells showed greater sphere-forming and higher cisplatin resistance. In this study, higher expression of CD133, YAP1, and ABCG2 was also associated with high CD24. A panel of CSC-related molecules were tested for their potential to detect bladder cancer, and the combination of CD24, CD49f, and NANOG showed promising sensitivity.

#### 3.5.4. Chemotherapy Sensitivity Testing

Several studies used organoids for chemosensitivity testing to investigate the feasibility of the approach for drug screening in drug development or even as a therapeutic decision tool. The response profile was compared with predicted drug response based on mutation background and with response in PDX models. Most of the studies investigated drug response in monoculture, except for the study by Kim et al. In this study, response to conventional therapy was decreased in assembloids in comparison with conventional tumor organoids [37].

The use of organoids derived from TURB samples as a chemosensitivity test to guide adjuvant therapy was investigated in one report, where Burges et al. evaluated low-grade, low-stage tumors sensitivity to intravesical drugs [29]. After 48 h in culture, spheroids were treated for 2 h, and response was evaluated using a proliferation agent and trypan blue. In this setting, the evaluability of clinically relevant cases (G1–G2) was 84% (21 out of 25 patients).

As a platform for drug discovery, single agents and combination therapies were evaluated in six successful organoids lines out of 15 primary TURB samples [36]. Similarly, response to cisplatin as a single agent or in combination with a G2/M checkpoint kinase inhibitor (WEE1 inhibitor, MK-1775) has also been evaluated in seven CTOS established from NMIBC [40].

As a proof of concept, drug testing was carried out in the organoid lines included in the biobank developed by Mullenders et al. and showed different chemosensitivity profiles among three selected organoid lines after a five days treatment [17].

Evidence for the clinical validity of drug testing using organoids was provided in the study by Lee et al., with organoid lines from chronologically distinct lesions from the same patient, before and after intravesical treatment, as well as from patients in the absence of additional treatment [15]. Treatment of the organoids with a wide range of chemo and targeted therapies revealed striking similarities and differences between the different organoid lines and partial correlation with their mutation profiles. Chemosensitivity was correlated with tumor progression in MIBC cases and recurrent NMIBC. Additionally, differences in drug response of metachronous organoid lines reflected changes in drug response of the primary tumor after treatment failure, whereas organoids from metachronous tumors without treatment showed similar response profiles. The drug response profiles observed in organoid culture were recapitulated in PDX models in three selected lines [15]. Concordant profiles between organoids and PDX models were also reported by Kita et al. [36], Amaral et al. [39], and Cai [16].

An advanced system for drug screening was developed by Gheibi et al., who made a microfluidic system to grow cancer spheroids after initial CTOS formation and embedding in Matrigel [32]. Unlike in standard spheroid methods, the spheroids in microfluidic chambers showed a constant growth over 30 days without development of necrosis. A distinct feature of this system is that it allowed drug response and relapse studies because the flow of media removes the dead cells. The drug response profile of the spheroids was in agreement with tumor response observed in the primary PDX models.

Different measures were used to evaluate drug response, including the CellTiter-Glo 3D assay [15,17,39], CellTiter-Blue [19], WST 8 assay [36], trypan blue [29], and variations in volume [32,40]. The starting time and duration of treatment also varied according to the seeding methods and drugs under investigation (Appendix A).

#### 3.5.5. Immunotherapy Testing

T cell cytotoxic activity on tumor cells was investigated in two reports using distinct methodologies. Neal et al. established cohort patient-derived tumor organoids using an air–liquid interface method from 100 individual patient tumors representing 19 distinct tissue sites, including bladder cancer [27]. In this setting, tumor organoids retained tumor epithelium and stroma, including immune cells, allowing functional tests of immunotherapeutic agents within a seven-day timeframe. In some tumors, including one bladder cancer organoid model, the treatment induced tumor cytotoxicity in parallel with tumor-infiltrating lymphocytes expansion and activation. This report showed that such an approach can be used to model the tumor immune microenvironment and response to immune checkpoint inhibitors [27]. The cytotoxic effect of activated T cells was also evaluated using co-cultivation of tumor organoids and engineered T cells targeting MUC1, an antigen highly expressed in both primary tumors and their derived organoids [41]. This system allows personalized preclinical CAR-T cell testing in bladder cancer based on patient-derived organoids.

### 3.6. Genetic Alterations in Bladder Cancer Organoids

Several publications reported that genomic alterations were highly conserved between the tissue–organoid pairs [15,16,17,33,41]. Table 3 shows the proportion of the reported mutations in each study in comparison with the two reference cohorts of NMIBC [48] and MIBC [49] with data extracted from cBioPortal. In addition to the studies reported in the table, targeted sequencing of the most common point mutation in *FGFR3* (S249) identified mutations in 2 out of 29 cases, which represented 67% (2/3) of the NMIBC cases. TP53 mutations identified after nutlin treatment were found in 22% of the cases (*n* = 36), mostly among MIBC [17]. Although there is a relatively low number of organoid lines with mutation data, the proportion of models showing mutations in driver genes agrees with the expected profile in bladder tumors. Thus, there seems to be no enrichment for particular genomic alterations in the organoid culture.

## 4. Discussion

Research spanning over three decades has shown that growing cells in three-dimensional (3D) cultures reduces the gap between cell cultures and live tissue, with improved physiological relevance and suitability for cell-based drug and toxicity screening [52]. Human tissue-derived organoids can be efficiently established both from normal bladder and from UC. Compared with traditional generation of cell lines, organoid cultures have higher efficiency and can be passaged for longer periods of time. Innovations in sample handling, culture systems, and imaging techniques allowed the development of an increasing number of multicellular models being used to study tumor biology, cell adhesion, cell migration, plasticity, and drug response [7,53].

Cancer stem cells (CSCs) are defined by their ability to self-renew and differentiate, which allows them to produce heterogeneous lineages of cancer cells within a tumor [54]. These cells can only be defined experimentally by their ability to generate a continuously growing tumor. It is believed that CSCs can be derived from normal stem cells after acquisition of mutations in driver genes, or from progenitor cells that acquire stem-like properties through multiple mutagenic events [54]. The cell surface markers necessary to identify CSCs, and in particular urothelial CSCs, are still a matter of debate. Putative cancer CSC populations have been described in the reports included in this analysis. A variety of markers were suggested to identify normal urothelial stem cells, including CD34, CD117 [30], and CD44 [17]. As for the identification of CSCs, expression of OCT3/4, CD133, and nestin [31], as well as CD24, CD49f, and NANOG [34] was proposed. The fundamental properties of self-renewal, the ability to differentiate into various cell types, the generation of organoids with different cell phenotypes, and formation of tumors after transplantation of organoids into immunocompromised mice offer compelling proof that the organoid systems described here enable the sustained CSC behavior. However, some studies also noted that, in some instances, the organoids stopped growing after a week or after a few passages, suggesting absence of cells with stemness abilities or absence of the right conditions for their proliferation.

Several studies reported establishment of organoid lines from PDX tumors as well as establishment of PDX models from organoids. The high efficiency of this interconversion allows the complementary use of both model systems for biological and drug-testing studies, including validation studies. Higher success rates of organoid establishment from PDX tumors in comparison to organoids derived directly from MIBC patient samples could be explained by the selection of aggressive clones during establishment in mice, or the increased amount of tissue and viable cells to work with. The results from the studies included in this review suggest that PDX-derived organoids can provide a valuable alternative as source material.

Some studies showed that tumor cells can adopt a basal phenotype over time when in organoid culture. This could be due to changes in environment or lack of factors in the medium; however, future studies will be necessary to further understand this phenomenon. The growth media formulations used to grow bladder cancer organoids have been adapted from studies with other tumor types. The effect of some of the currently used factors, such as the Wnt/β-catenin and MAPK modulators, noggin, and hepatocyte growth factor, remains to be established in bladder cancer. As we discover more about the clues necessary to grow organoids, preserve the different cell populations, and improve the biomimicity of the models, more-tailored growth conditions are likely to emerge. Standardization of sample handling and culture conditions is also a prerequisite for more-comparable drug screening.

Molecular classification of organoid lines should take into consideration the lack of stromal and immune signals, as well as increased proliferation stimulated by the added growth factors [15]. Thus, tumor cell phenotype-driven classification systems are preferred in this setting. The application of the consensus classifier [55] in future studies is highly encouraged to improve comparability between different studies and events of subtype plasticity.

Organoid biobanks capturing the genetic diversity and phenotype of patient tumors are promising tools for disease modeling and therapeutic screening. The establishment of a living biobank of bladder cancer organoids or pan-cancer organoid models including bladder has been reported by at least four studies [15,17,33,44]. These platforms considerably expand the options for high-throughput drug screening and development of targeted therapies. Additionally, drug response and molecular data obtained in the Lee et al. study, together with TCGA data, were used to identify biomarkers predictive of drug response in bladder cancer patients using network-based machine learning [21]. This study also noted that patient-derived organoid data, even in this limited setting, was better for predicting cisplatin response than the use of large-scale data from cancer cell lines. These results highlight the capacity of organoid models to reflect drug treatment outcomes in patients, and suggests that new insights may be gained from in silico analysis of the increasing amount of available high-quality data derived from organoids.

Organoid models have shown to be amenable for cryopreservation and genetic manipulation. Furthermore, the ability to generate independent organoid lines from individual tumors makes them an appealing tool to study tumor heterogeneity. However, genetic evolution and phenotypic shifts are still a concern, as some of the studies demonstrated selection of aggressive clones or acquisition of de novo genetic lesions.

Spheroid size affects cell behavior and function, as well as drug penetrance and cytotoxic response [56]. More advanced spheroid formation methods make use of technologies such as microchambers and bioprinting to improve control over spheroid size, cellular composition, and throughput. These technologies have proven to be useful in the generation and maintenance of organoid culture over longer periods of time without necrosis, and for development of organoid systems in which epithelial cells are combined with other stromal components.

From a clinical perspective, it is possible that organoids could be utilized in guiding alternative treatment selection, including immunotherapy, for patients unresponsive to established treatments in the metastatic setting, or for patients with recurring high-risk NMIBC tumors unfit for radical surgery. The fact that bladder cancer has a tendency for synchronous multifocality and that patients with recurring NMIBC often undergo multiple resections also allows for the study of tumor evolution and heterogeneity. Drug response to both standard and/or repurposed drugs for intravesical or systemic application in patient-derived organoid models can allow the identification of new therapeutic and treatment decisions, as described for prostate cancer organoids [57].

## 5. Conclusions

Lack of well-characterized and easily expandable tumor models has up to now hampered the development of targeted therapies and precision medicine for bladder cancer [58]. Bladder cancer organoids show high genetic and phenotypic concordance with the primary tissues and have emerged as a tool to more faithfully recapitulate tumor biology in preclinical research, as a platform for drug discovery, and as a potential guide for precision medicine in the future. Advances in bioengineering technology, such as microfluidic devices, bioprinters, and imaging, are likely to further standardize and expand the use of organoids.

## Figures and Tables

**Figure 1 cancers-14-02062-f001:**
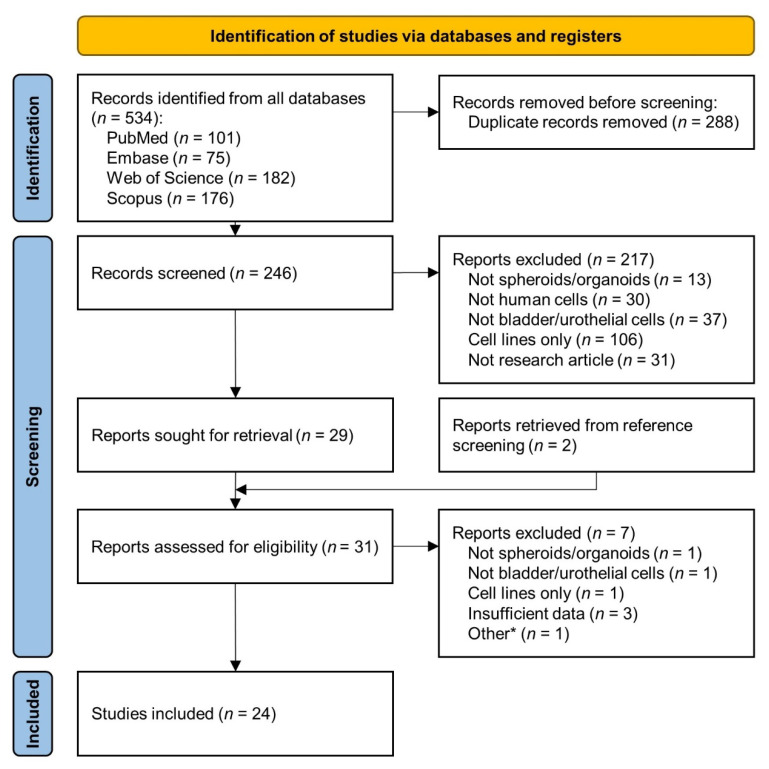
PRISMA flow diagram showing the study selection process. * Other indicates a report of network-based machine learning applied to published data generated in one of the studies included.

**Figure 2 cancers-14-02062-f002:**
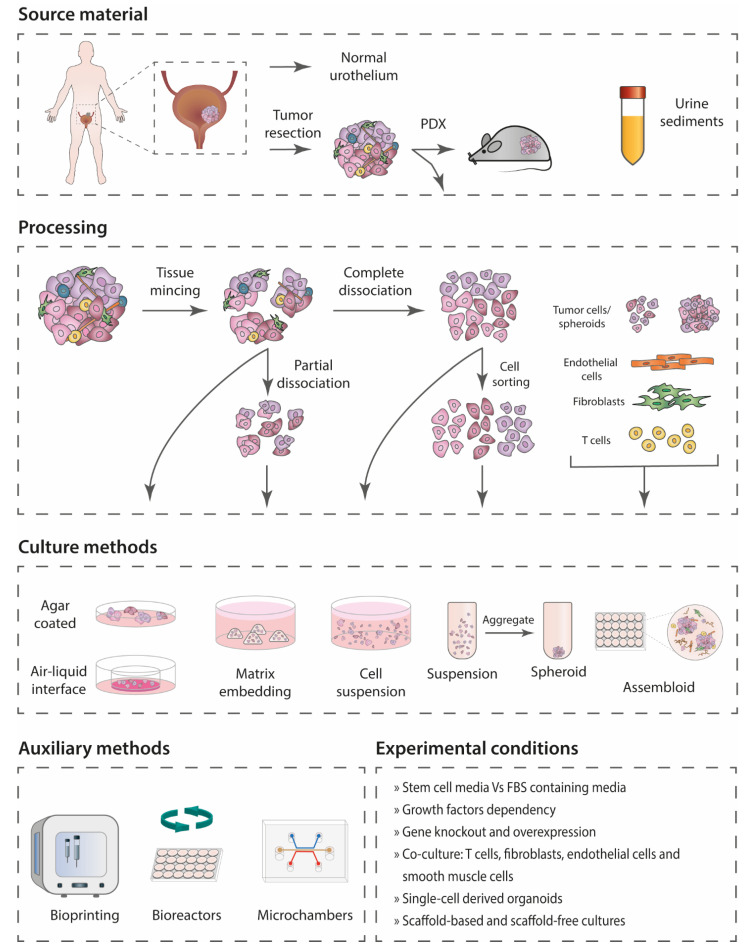
Overview of the different methods used for establishment of organoid models from normal urothelium and bladder cancer cells. PDX: patient-derived xenografts.

**Figure 3 cancers-14-02062-f003:**
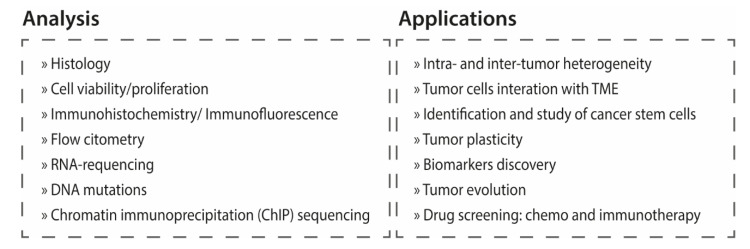
Summary of the main analysis used to characterize the organoids and compare with the primary tumors, and main applications of the models.

**Table 1 cancers-14-02062-t001:** Origin of the samples used in reviewed studies.

Paper	Sample Acquisition	Organoids/No. of Samples	Country
Burgues 2007 [29]	TURB	31/40	Spain
Fierabracci 2007 [30]	TUR-normal bladder	6/6	Italy
Bentivegna 2010 [31]	TURB	29/40 ^(a)^	Italy
Hofner 2013 [25]	PDX	2/2	Germany
Okuyama 2013 [22]	TURB, RC, PDX	128/152 ^(b)^	Japan
Yoshida 2015a [23]	surgical resection	119/176 ^(b)^	Japan
Yoshida 2015b [24]	PDX, resected, urine	NA ^(b),(c)^	Japan
Gabig 2016 [26]	TURB	na	USA
Gheibi 2017 [32]	PDX, patients	6	USA
Pauli 2017 [33]	Surgical resection	8/24 ^(d)^	USA
Ooki 2018 [34]	PDX	2	USA
Lee 2018 [15]	TURB, PDX	12/18	USA
Yoshida 2018 [35]	TURB	4	USA
Neal 2018 [27]	surgical resection	NA ^(e)^	USA
Kita 2019 [36]	TURB, PDX	6/15, 2(PDX)	Japan
Mullenders 2019 [17]	TURB, RC-normal and tumor	77/133 ^(f)^	Netherlands
Kim 2020 [37]	TURB, RC-normal and tumor	9	South Korea
Whyard 2020 [19]	TURB	4 ^(g)^	USA
Namekawa 2020 [38]	TURB	2	Japan
Yoon 2020 [28]	TURB	2	South Korea
Amaral 2020 [39]	PDX	2	USA
Murakami 2021 [40]	TURB, PDX	7	Japan
Yu 2021 [41]	RC	3	China
Cai 2021 [16]	PDX	2 (early & late P)	USA

^(a)^ 50 samples were collected, but only 40 had enough tissue for further processing. ^(b)^ Organoids from the same cohort of patients, here considered as one expanded cohort. ^(c)^ Missing or unclear information about number of samples. ^(d)^ Large cohort of pan-cancer organoids. Bladder and ureter organoids reported together. ^(e)^ Large cohort of pan-cancer organoids, at least one from urothelial carcinoma. ^(f)^ A total of 133 tissue samples were collected from 53 patients including normal-appearing urothelium and one or more tumor pieces per patient. ^(g)^ Two cultures, S2 and S2-1, were derived from the same line. Counted as four primary samples instead of five. TURB: transurethral resection of the bladder; RC: radical cystectomy; PDX: patient-derived xenografts; NA: information not available.

**Table 2 cancers-14-02062-t002:** Sample processing and culture conditions.

Study	Dissociation (Time, Min)	Strainer	Fraction	Matrix	Base Medium	Growth Factors, Other
Burgues 2007 [29]	M	no	NA	no	DMEM	FBS, L-glut, neaa
Fierabracci 2007 [30]	E (NA)	no	fragments	no	DMEM/F12	INS, Tf, PGT, putrescine, sodium selenite, β-met, bFGF, EGF
Bentivegna 2010 [31]	M&E (120–180)	40	FT	no	Adv DMEM/F-12	EGF, bFGF with or without FCS
Hofner 2013 [25]	E (60–120)	40	FT	no	DMEM/F-12	Refer to the article
Okuyama 2013 [22]	M&E (120)	NA	retained on strainer	Cellmatrix	DMEM/F12	^(a)^ (Glutamax, BSA, β-met, bFGF), HRG, activin A, long-IGF
Yoshida 2015a [23]	M&E (NA)	100, 40	retained on 100 or 40	no	DMEM/F12	^(a)^ (Glutamax, BSA)
Yoshida 2015b [24]	M&E (NA)	100, 40	retained on 100 or 40	Collagen I	DMEM/F12	^(a)^ (Glutamax, BSA)
Gabig 2016 [26]	M&E (120)	NA	retained on strainer	Matrigel	DMEM/F12	^(a)^ (Glutamax, BSA, β-met, bFGF), HRG
Gheibi 2017 [32]	M&E (25)	180, 40	retained on 40	Matrigel	RPMI	B27, EGF, bFGF or 30%FBS (PDX)
Pauli 2017 [33]	M&E (NA)	no	pellet	Matrigel	Adv DMEM/F-12	Glutamax, B27, Nac, NAM, EGF, FGF10, bFGF, A83-01, R-spondin, Noggin, PGE2, SB202190, ROCKi
Ooki 2018 [34]	M&E	NA	High-CD24, Low-CD24	no	DMEM/F-12	B27, EGF, bFGF
Lee 2018 [15]	M&E (15, 4)	100	FT	Matrigel	hepatocyte medium	Glutamax, EGF, ROCKi, FBS
Yoshida 2018 [35]	M&E (NA)	no	NA	no	DMEM/F-12	Glutamax, BSA, β-met
Neal 2018 [27]	M	no	all	Collagen I	Adv DMEM/F-12	Glutamax, HEPES, B27, Nac, NAM, A83-01, R-spondin, Noggin, EGF, Gastrin, SB202190, Wnt3a
Kita 2019 [36]	M&E (NA)	no	NA	Matrigel	As in Yoshida2018	As in Yoshida2018
Mullenders 2019 [17]	M&E (60)	70	FT	BME	Adv DMEM/F-12	B27, Nac, NAM, A83-01, FGF2/7/10, ROCKi
Kim 2020 [37]	M&E (60)	100	FT	Matrigel	Adv DMEM/F-12	Glutamax, HEPES, B27, Nac, NAM, A83-01, EGF, ROCKi
Whyard 2020 [19]	M&E (120)	40	retained on 40	BME	Adv DMEM/F-12	B27, Nac, NAM, A83-01, FGF2/7/10, HER3, ROCKi
Namekawa 2020 [38]	M&E (60)	100	FT	no	DMEM/F-12	^(a)^ (Glutamax, BSA, bFGF), ROCKi
Yoon 2020 [28]	M&E (60)	100	FT	Matrigel	Adv DMEM	Glutamax, HEPES, B27, Nac, NAM, A83-01, EGF, ROCKi
Amaral 2020 [39]	M&E (40–60)	100	FT	Matrigel	RPMI-1640	FBS, L-glut, neaa
Murakami 2021 [40]	E (75)	100, 40	retained on 100 or 40	Matrigel	DMEM/F-12	Glutamax, StemPro, BSA, β-met
Yu 2021 [41]	M&E (50)	70	FT	Matrigel	Adv DMEM/F-12	Glutamax, HEPES, B27, Nac, NAM, A83-01, R-spondin, Noggin, EGF, FGF2/10, SB202190
Cai 2021 [16]	M&E (65)	40	FT	Matrigel	Adv DMEM/F-12	B27, NAC, A83-01, R-spondin, Noggin, EGF, ROCKi

M: mechanical dissociation; E: enzymatic dissociation; FT: flow-through; FBS: fetal bovine serum; neaa: nonessential amino acids; EGF: epidermal growth factor; INS: insulin; Tf: transferrin; PGT: progesterone; β-met: 2-mercaptoethanol; bFGF: basic fibroblast growth factor; HGF: hepatocyte growth factor; Nac: N-acetylcysteine; NAM: nicotinamide; ROCKi: rho kinase inhibitor (Y-27632); HRG: heregulinB1.^(^^a)^ StremPro hESC SFM components, unclear from the reports if additional supplements were added.

**Table 3 cancers-14-02062-t003:** Mutation profile of the organoid lines in comparison with reference cohorts for non-muscle invasive bladder cancer (NMIBC) [48] and muscle-invasive bladder cancer (MIBC) [49] extracted from the cBioPortal [50,51].

	TCGA		Pauli2017	Lee2018		Yu2021	Cai2021
Mutations	NMIB	MIB	Tumor	NMIB	MIB	MIB	MIB
TP53	21%	48%	33% (1/3)	31% (5/16)	50% (3/6)	33% (1/3)	100% (1/1)
KMT2D	24%	28%	33% (1/3)	38% (6/16)	33% (2/6)	-	100% (1/1)
KDM6A	48%	26%	-	63% (10/16)	50% (3/6)	-	-
ARID1A	29%	25%	-	19% (3/16)	50% (3/6)	33% (1/3)	-
PIK3CA	28%	22%	33% (1/3)	31% (5/16)	33% (2/6)	-	-
KMT2C	11%	19%	-	44% (7/16)	33% (2/6)	33% (1/3)	-
RB1	4%	18%	33% (1/3)	6% (1/16)	0% (0/6)	33% (1/3)	-
EP300	14%	15%	-	6% (1/16)	33% (2/6)	33% (1/3)	100% (1/1)
FGFR3	45%	14%	33% (1/3)	56% (9/16)	50% (3/6)	-	-
STAG2	21%	14%	-	19% (3/16)	17% (1/6)	-	-
FAT1	26%	12%	-	-	-	33% (1/3)	100% (1/1)
CREBBP	21%	12%	33% (1/3)	19% (3/16)	33% (2/6)	33% (1/3)	
ERBB2	18%	12%	-	13% (2/16)	0% (0/6)	33% (1/3)	100% (1/1)
KMT2A	9%	11%	33% (1/3)	-	-	-	-
ERBB3	11%	10%	-	13% (2/16)	0% (0/6)	-	-
CDKN1A	11%	9%	-	0% (0/16)	17% (1/6)	33% (1/3)	-
FBXW7	14%	8%	-	6% (1/16)	33% (2/6)	-	-
TSC1	11%	8%	-	31% (5/16)	17% (1/6)	-	-
NFE2L2	6%	6%	-	0% (0/16)	17% (1/6)	-	-
RXRA	na	6%	33% (1/3)	-	-	-	-
RHOB	na	6%	33% (1/3)	-	-	-	-
CTNNB1	9%	3%	-	25% (4/16)	17% (1/6)	-	-
FOXA1	6%	3%	-	0% (0/16)	17% (1/6)	-	-
**Deletions/amplifications**							
CDKN2A	16%	33%	33% (1/3)	75% (12/16)	33% (2/6)	-	-
E2F3	3%	16%	33% (1/3)	19% (3/16)	33% (2/6)	-	-
CCND1	7%	12%	-	0% (0/16)	17% (1/6)	-	-
CCNE1	na	11%	-	13% (2/16)	0% (0/6)	-	-
MDM2	7%	9%	33% (1/3)	6% (1/16)	0% (0/6)	-	-
PTEN	na	5%	33% (1/3)	6% (1/16)	0% (0/6)	-	-
EGFR	na	5%	-	6% (1/16)	0% (0/6)	-	-

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
