# Peer review of "Patient-Derived Bladder Cancer Organoid Models in Tumor Biology and Drug Testing: A Systematic Review"

_cancers, 2022, doi:10.3390/cancers14092062_

Round 1

Reviewer 1 Report

Major targets and effects are missing in Table 1. Origin of the samples used in reviewed studies.

3.5.1. Functional studies, needs to be improved and add some mechanistic figure.

Also, please describe the mechanism of Chemotherapy sensitivity testing in general.

Author Response

We would like to thank the reviewers for the feedback and the opportunity to clarify and supplement the manuscript with figures and new perspectives. Please find bellow point by point reply to each comment and a description of the changes made in the manuscript. Revisions to the manuscript are marked up using the “Track Changes”. 

Comments reviewer 1:

  1. Major targets and effects are missing in Table 1. Origin of the samples used in reviewed studies.

R: It is unclear what the reviewer is referring to. The origin of the samples is stated in table 1 column “Country”. We hope that the new added figure helps clarify the major types and applications of the organoids.

  1. 5.1. Functional studies need to be improved and add some mechanistic figure.

R: Two figures have now been added to the manuscript to summarize the sample-processing and different culture conditions, methods for analysis and applications.

  1. Also, please describe the mechanism of Chemotherapy sensitivity testing in general.

R: The mechanism of chemotherapy testing is described in text, Line 462-465 “Different measures were used to evaluate drug response including the CellTiter-glo 3D assay [15,17,39], CellTitter-Blue [19], WST 8 assay [36], trypan blue [29] and variations in volume [32,40]. The starting time and duration of treatment also varied according to the seeding methods and drugs under investigation (Supplementary Table 2).”

Considering the variation of methods and drugs used in different papers the tables and text seem to be the best way to provide a complete overview of the information.

Reviewer 2 Report

This meta-analysis is summarizing patient-derived BC organoid models. The manuscript is well written and the topic of discussion is interesting since there is a gap in understanding the tumor pathogenesis owing to its heterogeneity and thus limiting the available treatment options. I have a few comments that may improve the article's readability:

1- Summaraizing figures are needed to elaborate several paragraphs over the article such as different spherical cancer models and tissue dissociation techniques.

2- Abbreviations in any figures need to be explained in the legends separately from the main text.

3- The article has lots of information. However, only 52 references only are included!

4- Sections of comparison with primary tumors and chemotherapy sensitivity testing need to be summarized.

Author Response

We would like to thank the reviewers for the feedback and the opportunity to clarify and supplement the manuscript with figures and new perspectives. Please find bellow point by point reply to each comment and a description of the changes made in the manuscript. Revisions to the manuscript are marked up using the “Track Changes”. 

Comments reviewer 2:

This meta-analysis is summarizing patient-derived BC organoid models. The manuscript is well written and the topic of discussion is interesting since there is a gap in understanding the tumor pathogenesis owing to its heterogeneity and thus limiting the available treatment options. I have a few comments that may improve the article's readability:

  1. Summarizing figures are needed to elaborate several paragraphs over the article such as different spherical cancer models and tissue dissociation techniques.

R: We agree that a summarising figures can be useful, and 2 figures were added: Figure 2 depicting the different methodological approaches has now been included in the section 3.3., and Figure 3. presenting an overview of the analysis used to characterize the organoids and main applications of the models.

  1. Abbreviations in any figures need to be explained in the legends separately from the main text.

R: Thanks for highlighting that, the abbreviation list after each table and figure have now been updated.

  1. The article has lots of information. However, only 52 references only are included!

R: the revised version contains 55 references. The citations seem adequate to the content.

  1. Sections of comparison with primary tumors and chemotherapy sensitivity testing need to be summarized.

R: A closing paragraph was added to the section 3.5.1 Comparison with primary tumors as follows: “Overall, the existing data suggest that tumor cell morphology and genetic features are preserved in organoid culture. The gene expression profile of the organoids also resembles the parental tumors in terms of molecular subtype and cellular composition. However, in some instances the gene expression profile changed in culture for some of the organoids, particularly those derived from tumors with luminal phenotype, such changes were reversible.”

Reviewer 3 Report

To the authors of the manuscript “Patient-derived bladder cancer organoid models in tumor biology and drug testing: a systematic review”.

This review is an important tool for any researcher who thinks about working with organoids, from their applicability, to the culture media and tests that can be carried out based on our research, without forgetting to mention the problems and limitations that working with organoids can entail.

The tables included in the manuscript as well as the supplementary information are very useful. And the references are current and specific to bladder cancer for the most part.

I would review some unfortunate English expressions and abbreviations like “bladder cancer”. Otherwise it is a very comprehensive review with strong exclusion criteria.

Author Response

We would like to thank the reviewers for the feedback and the opportunity to clarify and supplement the manuscript with figures and new perspectives. Please find bellow point by point reply to each comment and a description of the changes made in the manuscript. Revisions to the manuscript are marked up using the “Track Changes”. 

Comments reviewer 3:

To the authors of the manuscript “Patient-derived bladder cancer organoid models in tumor biology and drug testing: a systematic review”.

This review is an important tool for any researcher who thinks about working with organoids, from their applicability, to the culture media and tests that can be carried out based on our research, without forgetting to mention the problems and limitations that working with organoids can entail.

The tables included in the manuscript as well as the supplementary information are very useful. And the references are current and specific to bladder cancer for the most part.

I would review some unfortunate English expressions and abbreviations like “bladder cancer”. Otherwise it is a very comprehensive review with strong exclusion criteria.

Reply: We thank the reviewer for taking the time to review and give feedback on our manuscript. The manuscript has now been revised to correct and improve the text. Revisions to the manuscript are marked up using the “Track Changes”.

Reviewer 4 Report

This study was described the utility of patient-derived bladder cancer organoid models for bladder cancer. The reviewer thinks that this study is unique and very interesting. The reviewer would like to suggest some critiques as follows.

Minor

  1. On line 17, “with not infrequently poor outcome” is unclear. The authors should revise this point.
  2. On line 29, what is PDX models?
  3. On line 38, non-muscle invasive tumors and muscle invasive tumors were unclear. In addition, what is “~75%”?
  4. On line 40, what is metastasis? Local recurrence? Distant metastasis?
  5. On line 43, the authors should quote with regard to the 5-year survival. Overall? Cancer-specific?
  6. On line 67, why the authors change the font? For example, “multicellular tumor spheroids”.

Author Response

We would like to thank the reviewers for the feedback and the opportunity to clarify and supplement the manuscript with figures and new perspectives. Please find bellow point by point reply to each comment and a description of the changes made in the manuscript. Revisions to the manuscript are marked up using the “Track Changes”. 

Comments reviewer 4:

This study was described the utility of patient-derived bladder cancer organoid models for bladder cancer. The reviewer thinks that this study is unique and very interesting. The reviewer would like to suggest some critiques as follows.

Minor

  1. On line 17, “with not infrequently poor outcome” is unclear. The authors should revise this point.

R: Sentence was rephased and now reads as follows: “Bladder cancer is a common and highly heterogeneous malignancy with relatively poor outcome”

  1. On line 29, what is PDX models?

R: Abbreviation description is now included in the text and reads as follows: “Organoids sensitivity to chemotherapy matched the response in patient derived xenograft (PDX) models and predicted response based on clinical and mutation data.”

  1. On line 38, non-muscle invasive tumors and muscle invasive tumors were unclear. In addition, what is “~75%”?

R: The sentence was rephrased and now read as follows: “Bladder cancer is a common and highly heterogeneous malignancy that manifests in two major patterns, i.e. as non-muscle invasive tumors (NMIBC) consisting of around 75% of the new cases with a generally better prognosis but frequent relapses, and as muscle invasive tumors (MIBC) with high risk of regional and distant metastasis and poor prognosis.”

  1. On line 40, what is metastasis? Local recurrence? Distant metastasis?

R: the sentence was changed to include “regional and distant metastasis”, see above.

  1. On line 43, the authors should quote with regard to the 5-year survival. Overall? Cancer-specific?

R: sentence updated: “At this stage, despite the treatment, the 5-year overall survival is only 50% [2]”

  1. On line 67, why the authors change the font? For example, “multicellular tumor spheroids”.

R: the intension was to highlight and allow a quick grasp of key concepts, in this case the different organoid types and terminologies, but we understand that this can cause confusion instead. Bold font has now been removed.